# Suicidal and aggressive behavior among populations within institutional quarantine and isolation centers of COVID-19 in eastern Ethiopia: A cross-sectional study

**Tadesse Misgana** [1]*, **Dejene Tesfaye**[1], **Mandaras Tariku**[1], **Tilahun Ali**[1], **Daniel Alemu**[1], **Yadeta Dessie**[2]

1 Department of Psychiatry, College of Health and Medical Sciences, Haramaya University, Harar, Ethiopia,
2 School of Public Health, College of Health and Medical Sciences, Haramaya University, Harar, Ethiopia

☯ These authors contributed equally to this work.
* tadessemisgana25@gmail.com

## Abstract

### Introduction

The coronavirus disease is still not under the control globally and has caused various mental health problems such as depression, anxiety, suicide, and aggressive behavior in different populations. The pandemic-related issues which are applied to control the pandemic such as protection measures against COVID-19, social distancing, isolation, and quarantine can also trigger mental health problems.

### Objective

This study aimed to assess suicidal behavior and aggression, and its correlates during COVID-19 among populations within institutional quarantine and isolation centers in Ethiopia.

### Method

A cross-sectional study was conducted among a sample of 392 participants. The convenience sampling method was used to select the study participants. Suicide Behavioral Questionnaire-Revised (SBQ-R) and the Modified Overt Aggression Scale (MOAS) were applied to determine the suicide and aggressive behavior of study participants respectively. Epi-data 3.1 and SPSS 20.0 were used to enter and analyze the data respectively. Logistic and linear regressions were fitted to explore correlates associated with suicidal behavior and aggression respectively.

### Results

The prevalence of suicidal behavior was 8.7% (95% CI: 6.1, 11.5) whereas the mean total score of behavioral aggression was 2.45±5.90 (95% CI: 1.84, 3.08). Being female (AOR = 2.63, 95% CI: 1.09, 6.32), having common mental disorders (AOR = 6.08, 95% CI: 2.32,

**Data Availability Statement:** All relevant data are within the manuscript and its Supporting Information files.

**Funding:** This study was funded by Haramaya University (Grant No. HUCF-2020-02-NA-06).

**Competing interests:** The authors have declared that no competing interests exist.

15.93), manifesting the symptoms of COVID-19 (AOR = 2.17, 95% CI: 1.48, 2.86), and poor social support (AOR = 7.30, 95% CI: 1.44, 37.10) were significantly associated with suicidal behavior, whereas male gender (β coefficient = 3.0, 95% CI: 1.35, 4.70), low level of knowledge about COVID-19 (β coefficient = 1.87, 95% CI: 1.09, 3.41), and substance use (β coefficient = 1.7, 95% CI: 1.23, 6.47) were positively associated with mean overt aggression score.

## Conclusion

The present study revealed that suicidal and aggressive behaviors were prevalent with significant correlates. Therefore, it is important and required to provide focused mental health and psycho-social services for the selected and high-risk populations such as those in quarantine and isolation centers for being suspected.

## Introduction

The WHO declared COVID-19 as a pandemic a year ago [1] and has become a global concern [2]. Since the beginning, various countries have been struggling to mitigate the spread of the pandemic through different protection measures such as quarantine and isolation, massive lockdowns, and vaccines. Currently, there is no exact prediction for how long the pandemic will continue, how many people will be infected, or, consequently, how long people's lives will be troubled [3, 4]. As observed in preceding epidemics and pandemics, in addition to burdening the health system, the threat and insecurity surrounding public protection, which is difficult to predict, can have negative consequences on individuals' mental health like suicide and aggressive behavior [5, 6].

In addition to this, the protection measures taken against COVID-19 are applied to control the pandemic like mass quarantine and isolation, and consequently, the social and economic fallout can precipitate mental health problems [1, 7]. With such long-lasting suffering and severe condition, mental health problems can result in suicidal-related thoughts and behaviors [8, 9]. The pandemic could adversely affect other known precipitants of suicide [10]. For instance, intimate partner violence and drinking alcohol might rise during the period of lockdown [11].

A strong association has been seen between the increase in suicide and traumatic events such as the viral pandemic. In one systematic review study, suicidal-related behavior increased at the time of COVID-19 pandemic as compared to the pre-COVID-19 pandemic period [12]. Another recent study conducted in Iran during the COVID-19 Pandemic shows that the magnitude of suicidal thoughts was increased to 20.8% [13]. In earlier pandemics, similar conditions were reported [5]. The suicide-related consequences of the pandemic might vary depending on countries' public health control measures, socio-cultural and demographic structures, availability of digital alternatives to face-to-face consultation, and existing supports. The effects might be worse in resource-poor settings like Ethiopia where economic adversity is compounded by inadequate welfare support.

If the extended restriction of personal movement and disruption in daily life contribute to hindering individual goals, it would logically follow that aggression would increase during lockdowns [14]. In support of this, intimate partner aggression perpetration is associated with pandemic-related negative outcomes, such as social isolation [15] and economic stressors [16].

Taken together, this literature provides indirect support for the view that people who experience heightened COVID-19 stress are at greater risk for intimate partner aggression perpetration. However, no studies have yet directly addressed suicidal behaviors and levels of measured aggression as a function of lockdown status during the COVID-19 pandemic in Ethiopia. Therefore, this study aimed to assess suicidal behavior, aggression, and its correlates during the COVID-19 pandemic among populations within institutional quarantine and isolation centers.

## Materials and methods

### Study setting, design, and period

A multi-center cross-sectional study was done in the Harari regional state and eastern Hararghe zone of Oromia regional state, eastern Ethiopia and the participants were recruited from November 15 to December 31, 2020. Ethiopia has established a COVID-19 task force as part of the response to the pandemic. In addition to selected health facilities for the treatment of patients with symptomatic COVID-19, there were isolation centers that catered to the infected but asymptomatic individuals. Furthermore, there were designated quarantine centers for suspected cases and those awaiting test results.

### Population and eligibility

Suspected cases of COVID-19 aged 18 and above in quarantine and isolation centres were considered as the study population, and those who were unable to participate due to serious underlying health conditions or severe symptoms of COVID-19 were excluded.

### Sample size determination

The sample size was estimated using two approaches based on the objectives of the study. Accordingly, for each specific objective, the sample size is calculated separately and the larger sample size was taken to be used for this study. For the first objective, to estimate the prevalence of suicidal behavior, a single population proportion formula was used with the following assumptions: prevalence of suicidal behavior 50% since there is no study conducted during the COVID-19 pandemic, a 5% margin of error, and a 5% of non-response rate. The sample size was 422. The sample size for the second specific objective is determined by considering factors that are significantly associated with the suicidal behavior (being female), the two-sided confidence level of 95%, the margin of error of 5%, power of 80%, and the ratio of exposed to unexposed 1:1 using STAT CALC of Epi Info Version 7. Finally, the required sample size for this particular study is decided by taking the maximum from the calculated sample size, which is422. In this study, the latter sample size was considered to increase the power of the study.

### Sampling technique and procedure

From two isolation centers and 8 quarantine centers in Harari regional state, randomly we selected one isolation center and three quarantine centers. Again, the East Hararghe zone of Oromia regional state had one isolation center and 12 quarantine centers, and we randomly selected 4 of them. Finally, the participants were conveniently selected from each center proportionally based on the number of suspected cases they occupied. The data were collected in face-to-face by considering all the essential precautions recommended by WHO to control the transmission of COVID-19.

## Data collection instruments

The semi-structured questionnaire which contains 8 sections was used. The questionnaires for assessing the socio-demographic and clinical factors were adopted by reviewing the literature. The Self-Reporting Questionnaire (SRQ-20) was used to determine the presence of common mental disorders (CMD).The SRQ-20 was validated in Ethiopia at a general population [17]. Alcohol, Smoking, and Substance Involvement Screening Test (ASSIST) [18] and Oslo 3-item Social Support Scale (OSSS-3) [19] were administered to assess the substance use and social support of the participants respectively.

Suicide Behavioral Questionnaire-Revised (SBQ-R) was used to screen the suicidal behaviors of study participants. Each item assesses a different dimension of suicidality (or risk of suicide) and the higher the obtained score reflects the higher risk for subsequent suicidal behavior. SBQ-R is made up of 4 items scale ranging from 3 to 18, with a cutoff score of $\geq 7$ showing the presence of suicidal behavior [20]. In this study, the reliability of SBQ-R was checked and it was a kappa value of 0.91.

The Modified Overt Aggression Scale (MOAS) was used to screen the level of aggression among suspected cases at quarantine and isolation centers. MOAS has a total of 16 items with four categories that assesses aggression, namely verbal aggression, aggression against property, auto-aggression, and physical aggression. The severity of aggressive behavior ranges from 0 (no aggression) to 4 points (maximum violence) for each category or it means each domain is multiplied by the number of the category ranging from 0 (no aggression) to 40 (maximum aggression) [21].The MOAS has been tested at various centers in the developed world and found to be a valid indicator of the type and severity of aggression with good inter-rater reliability. The MOAS has excellent psychometric properties; the correlation coefficient between raters was 0.9. Furthermore, it is validated in Nigeria [22].

## Data quality control

The questionnaire was prepared first in the English language and translated to local languages (Afan Oromo, and Amharic languages) and back to the English language by professional translators to maintain the consistency of the data collection tool. Standardized tools specific to the research objectives were adopted and intensive training was given to the data collectors and supervisors. A pre-test was done on 5% of the sample size in Dire Dawa city administration in order to evaluate the acceptability and applicability of the procedures and tools. Data collectors were closely supervised during the data collection and two data clerks were doing the data entry to minimize errors during data entry.

## Data processing and analysis

The data were entered and analyzed using Epidata3.1 and SPSS 20.0 respectively. Absolute frequencies and percentages were calculated for categorical variables, and mean and standard Deviation were calculated for continuous variables. Whenever the data were not normally distributed, median and IQR were calculated. Bi-variable and multivariable logistic regression was carried out to determine associations of suicidal behavior and independent variables. Fourteen variables were included during bivariate analysis and seven variables with P-value $\leq 0.25$ was taken into consideration in the multivariable model to control for all possible confounders. Linear regression was performed for the total score of the behavioral aggression to investigate the association with different explanatory variables. In simple linear regression, twelve variables were included and seven were included in the final multiple regression model based on their significance level. The adjusted R2, the unstandardized B coefficient with its

95% CI, and the p-value were calculated. For both tests, the significance level was set at a p-value of 0.05.

## Ethics consideration

The study was conducted following the principles of the Declaration of Helsinki. Ethical clearance was obtained from the Institutional Health Research Ethics Review Committee (IHRERC) of the College of Health and Medical Sciences, Haramaya University with the approval number of IHRERC/243/2020. Written informed consent was obtained from the participants and those identified as high risk of suicidal and aggressive behavior were referred to a psychiatry specialty clinic for advance assessment and management.

## Results

### Socio-demographic descriptions

Of 423 participants, the response rate was 92.7% (392 participants). About 205 (52.3%) were males, with a median age of 30.5(IQR = 10.6) years. About 204 (52%) of the participants were married and 39 (9.9%) indicated they could not read and write (Table 1).

### Clinical descriptions

About 2.0% had a previous history of known mental illness and 18.9% didn't put into practice any COVID-19 control measures. In another way, 227 (57.9%) of the participants know how to protect themselves from the disease while 193 (49.2%) know how it is transmitted. About 101 (25.8%) participants had inadequate accessibility to personal protective equipment and 205 (52.3%) heard about COVID-19 twice daily. One-third (27%) of the participants manifested the typical symptoms of COVID-19 (Table 2).

**Table 1. Distribution of socio-demographic characteristics of populations within institutional quarantine and isolation centers of COVID-19 in eastern Ethiopia, 2020 (N = 392).**

| Variables | Frequency | Percentage (%) |
|---|---|---|
| **Sex** | | |
| Male | 205 | 52.3 |
| Female | 187 | 47.7 |
| **Age (years)** | | |
| <30 | 244 | 62.2 |
| 30–39 | 81 | 20.7 |
| 40–49 | 47 | 12.0 |
| ≥50 | 20 | 5.1 |
| **Marital status** | | |
| Never married | 179 | 45.7 |
| Married | 204 | 52.0 |
| Others* | 9 | 2.3 |
| **Educational status** | | |
| Can't write and read | 39 | 9.9 |
| Only able to write and read | 74 | 18.9 |
| Primary school | 63 | 16.1 |
| Secondary and preparatory school | 101 | 25.8 |
| College and above | 115 | 29.3 |

Others* = divorced, widowed, separated

**Table 2. Distribution of clinical characteristics of populations within institutional quarantine and isolation centers of COVID-19 in eastern Ethiopia, 2020 (N = 392).**

| Variables | Frequency | Percentage (%) |
|---|---|---|
| **History of mental illness** | | |
| Yes | 8 | 2.0 |
| No | 384 | 98.0 |
| **History of chronic medical illness** | | |
| Yes | 25 | 6.4 |
| No | 367 | 93.6 |
| **Practicing COVID-19 control measures** | | |
| Yes | 74 | 18.9 |
| No | 318 | 81.1 |
| **Accessibility of PPE** | | |
| Yes | 291 | 74.2 |
| No | 101 | 25.8 |
| **COVID-19 information** | | |
| None | 71 | 18.1 |
| 2 times daily | 205 | 52.3 |
| 3 and more times daily | 116 | 29.6 |
| **Symptoms of COVID-19** | | |
| Yes | 106 | 27.0 |
| No | 286 | 73.0 |
| **Perceived knowledge about COVID-19** | | |
| Poor | 130 | 33.2 |
| Good | 262 | 66.8 |

## Psycho-social and substance use descriptions

About 16.5%, 86.5%, and 16.5% of the respondents were reporting that they used tobacco, khat, and alcohol in the previous three months respectively, whereas 165 (42.1%) had received good social support. About 13.5% of the participants had common mental disorders as screened using SRQ-20 (Table 3).

## Suicidal behavior

The reported prevalence of suicidal behavior among suspected cases of COVID-19 was 8.7% (95% CI: 6.1, 11.5) (SBQ-R≥7). The lifetime prevalence of suicidal ideation, plan, and attempt was 21.9%, 4.8%, and 2.3%, respectively. The 12-monthprevalence of suicidal ideation was 8.0% (95% CI: 5.2, 10.7). The mean SBQ-R score was 3.88 with a standard deviation of 2.26.

## Behavioral aggression (The Modified Overt Aggression Scale)

According to the Modified Overt Aggression Scale (MOAS), the mean total score for behavioral aggression was 2.45±5.90 (95% CI: 1.84, 3.08). About 20.4% of the participants scored above the mean of the overt aggression scale. The 4 main categories were described in Table 4.

## Factors associated with suicidal behavior

Bi-variable logistic regression model revealed that suicidal-related behavior was significantly associated with age, sex, chronic medical illness, symptoms of COVID-19, common mental disorders (CMD), social support, and use of the substance. However during multivariable

**Table 3. Distribution of psycho-social and substance-related characteristics of populations within institutional quarantine and isolation centers of COVID-19 in eastern Ethiopia, 2020 (N = 392).**

| Variables | Frequency | Percentage (%) |
|---|---|---|
| **Social support** | | |
| Poor | 57 | 14.5 |
| Moderate | 170 | 43.4 |
| Strong | 165 | 42.1 |
| **Common mental disorders** | | |
| Yes | 53 | 13.5 |
| No | 339 | 86.5 |
| **Lifetime Alcohol use** | | |
| Yes | 79 | 20.2 |
| No | 313 | 79.8 |
| **Lifetime Tobacco use** | | |
| Yes | 71 | 18.0 |
| No | 321 | 82.0 |
| **Lifetime khat use** | | |
| Yes | 366 | 93.4 |
| No | 26 | 6.6 |
| **Current Alcohol use** | | |
| Yes | 65 | 16.5 |
| No | 327 | 83.5 |
| **Current Tobacco use** | | |
| Yes | 65 | 16.5 |
| No | 327 | 83.5 |
| **Current khat use** | | |
| Yes | 339 | 86.5 |
| No | 53 | 13.5 |

logistic regression analysis, female gender (AOR = 2.63, 95% CI: 1.09, 6.32), having CMD (AOR = 6.08, 95% CI: 2.32, 15.93), manifesting symptoms of COVID-19 (AOR = 2.17, 95% CI: 1.48, 2.86),and poor social support (AOR = 7.30, 95% CI: 1.44, 37.10) were significant associated with suicide (Table 5).

## Factors associated with behavioral aggression

During multiple regression analysis, male gender, knowledge about COVID-19, and substance use were significantly and positively associated with mean overt aggression score. The mean score of the overt aggression scale of male participants was increased by three units (95%CI: 1.35, 4.70) as compared to female participants. Additionally, the study participants who developed COVID-19 symptoms and had poor knowledge about COVID-19 had their overt

**Table 4. Description of the categories of behavioral aggression among populations within institutional quarantine and isolation centers of COVID-19 in eastern Ethiopia, 2020 (N = 392).**

| Categories | Mean ± Standard deviation | 95% CI of mean |
|---|---|---|
| **Verbal aggression** | 0.23±0.59 | 0.17, 0.29 |
| **Aggression toward objects** | 0.47±1.23 | 0.35, 0.59 |
| **Auto-aggression** | 0.81±2.06 | 0.62, 1.02 |
| **Physical aggression** | 0.93±2.91 | 0.65, 1.22 |

**Table 5. Bivariate and multivariate logistic regression analysis of factors associated with suicidal behavior among populations within institutional quarantine and isolation centers of COVID-19 in eastern Ethiopia, 2020 (N = 392).**

| Study Variables | Suicidal behavior | | COR (95% CI) | AOR (95% CI) | | P-values |
|---|---|---|---|---|---|---|
| | **With suicidal behavior** | **Without suicidal behavior** | | | | |
| **Sex** | | | | | | |
| Male | 9 (4.4%) | 196 (95.6%) | 1 | 1 | | |
| Female | 25 (13.4%) | 162 (86.6%) | 3.36 (1.53, 7.40) | 2.63 (1.09, 6.32) | | 0.03 |
| **Age (in years)** | | | | | | |
| <30 | 16 (6.6%) | 228 (93.4%) | 1 | 1 | | |
| 30–39 | 8 (9.9%) | 73 (90.1%) | 1.56 (0.64, 3.79) | 1.19 (0.40, 3.53) | 0.74 | |
| 40–49 | 6 (12.8%) | 41 (87.2%) | 2.08 (0.77, 5.64) | 1.64 (0.49, 5.41) | 0.41 | |
| ≥50 | 4 (20.0%) | 16 (80%) | 3.56 (1.07, 11.91) | 2.49 (0.59, 10.39) | 0.21 | |
| **Chronic medical illness** | | | | | | |
| No | 28 (7.6%) | 339 (92.4%) | 1 | 1 | | |
| Yes | 6 (24.0%) | 19 (76.0%) | 3.82 (1.41, 10.35) | 2.23 (0.57, 8.67) | 0.24 | |
| **Symptoms of COVID-19** | | | | | | |
| No | 18 (6.3%) | 268 (93.7%) | 1 | 1 | | |
| Yes | 16 (15.1%) | 90 (84.9%) | 2.65 (1.29, 5.41) | 2.17 (1.48, 2.86) | 0.01 | |
| **Common mental disorders** | | | | | | |
| **No** | 17 (5.0%) | 322 (95.0%) | 1 | 1 | | |
| **Yes** | 17 (32.1%) | 36 (67.9%) | 8.94 (4.20, 19.03) | 6.08 (2.32, 15.93) | <0.001 | |
| **Substance use** | | | | | | |
| No | 21 (7.0%) | 281 (93.0%) | 1 | 1 | | |
| Yes | 13 (14.4%) | 77 (85.6%) | 2.25 (1.08, 4.71) | 1.89 (0.75, 4.74) | 0.17 | |
| **Social support** | | | | | | |
| Strong | 2 (3.5%) | 55 (96.5%) | 1 | 1 | | |
| Poor | 27 (16.4%) | 138 (83.6%) | 5.38 (1.24, 23.39) | 7.30 (1.44, 37.10) | 0.01 | |
| Moderate | 5 (2.9%) | 165 (97.1%) | 0.83 (0.16, 4.42) | 0.87 (0.14, 5.25) | 0.88 | |

Note: Hosmer lem show test = 0.72

**Abbreviation:** AOR, Adjusted odds ratio; COR, Crude Odds Ratio; CI, Confidence Interval; COVID-19, Corona Virus Disease 2019

aggression score increased by 2.5 units (95%CI: 1.26, 8.24) and 1.8 units (95%CI: 1.09, 3.41) respectively as compared with those who had no COVID-19 symptoms and had a good understanding. The average score of aggression was 1.7 (95% CI: 1.23, 6.47) higher for those who used substances (Table 6).

## Discussion

The current study revealed the suicidal and aggressive related thoughts and behavior of suspected cases of the COVID-19 pandemic in eastern Ethiopia. The prevalence of suicidal-related thoughts and behavior was 8.7%. Of this, 21.9%, 4.8%, and 2.3% had suicidal ideation, plan, and attempt respectively. The one-year prevalence was 18.0%. These findings were higher than a meta-analysis study conducted among 308,596 participants across 54 studies that reported the pooled prevalence of suicide ideation (10.81%) and attempts (4.68%) [23]. The study further demonstrates that the mean total score of behavioral aggression was found to be 2.45±5.90 and about one-fourth (20.4%) of the participants scored above the mean of overt aggression scale.

The current study reported that women were facing a higher suicidal burden than men during the COVID-19 pandemic. This finding was in line with studies done in Bangladesh [24],

**Table 6. Multiple linear regression analysis of factors associated with behavioral aggression among populations within institutional quarantine and isolation centers of COVID-19 in eastern Ethiopia, 2022.**

| Variables | β coefficient (95% CI) | P-values |
|---|---|---|
| **Sex** | | |
| Female | 0 | 0 |
| Male | 3.00 (1.35, 4.70) | <0.001 |
| **Age** | -0.01 (-0.12, 0.09) | 0.120 |
| **Marital status** | | |
| Married | 0 | 0 |
| Never married | -1.12 (-3.35, 1.12) | 0.430 |
| Divorced/widowed/separated | 3.90 (-0.25, 8.10) | 0.320 |
| **Educational status** | | |
| College and above | 0 | 0 |
| Secondary school | 1.30 (-1.08, 3.55) | 0.360 |
| Primary school | 0.55 (− 0.42, 1.52) | 0.412 |
| Able to write and read | 1.37 (-1.20, 3.90) | 0.810 |
| Unable to write and read | − 0.71 (− 1.34, 0.49) | 0.250 |
| **Presence of COVID-19 symptoms** | | |
| No | 0 | 0 |
| Yes | 2.50 (0.26, 4.24) | 0.14 |
| **Knowledge about COVID-19** | | |
| Good | 0 | 0 |
| Poor | 1.87 (1.09, 3.41) | 0.03 |
| **Social support** | | |
| Strong | 0 | 0 |
| Moderate | -0.03 (-0.02, 0.05) | 0.21 |
| Poor | 2.10 (0.27, 4.35) | 0.23 |
| **Substance use** | | |
| No | 0 | 0 |
| Yes | 1.71 (1.23, 6.47) | 0.03 |

NOTE: Model summary: $R^2$ = 93.2%, adjusted $R^2$ = 89.2%

Abbreviations: COVID-19, Corona Virus Disease 2019

Japan [25], and India [26]. Numerous potential explanations are there for this gender difference. The first one is domestic/gender violence. As shown in different countries, restrictions on movement, like quarantine, and isolation, increase the violence against women [27–29], which is identified to raise suicidal ideation and behaviors in women [30]. The findings of this study call for extra efforts to defend women from gender violence. Secondly, being without a job or a decline in job and a considerable raise in psychological distress may increase suicide rates [27]. United Nations Women has warned about the unequal increase in joblessness among women during the COVID-19 pandemic [31].

As supported by a study done in Bangladesh [12], individuals who had common mental disorders were 6 times more prone to develop suicidal-related behavior. Several psycho-pathological factors like depression, fears, stress, and insomnia will significantly intensify the risk of suicidal behavior [24, 32, 33], and about 90% of the suicide occurrences were attributed to mental disorders [34]. Therefore, to decrease suicides during the period of the COVID-19 pandemic, it is essential to reduce stress, anxiety, and aloneness.

Participants who had COVID-19 signs and symptoms were more likely to have suicidal behavior as compared to the participants who remains asymptomatic. The pandemic-

associated issues might have major credit to suicidality by aggravating the mental health problems. Therefore, there is a call for critical commencement of community-based awareness programs, facilitation of genuine, consistent, and updated information, and strict monetization of propaganda, misinformation, conspiracy theories, etc., related to the pandemic [35]. These strategies can boost community understanding and protective behaviors and reduce their fear regarding COVID-19; as a result, decrease suicidality.

Having less social support was also a predictor of suicidal thoughts and behavior. This result was consistent with other studies [36–38]. The likely reason is that suspected individuals who were in centers are more likely to contract the disease and their close family and friends may be concerned about being infected, resulting in relatively less support. This may lead to suicide [39]. Therefore, close consideration should be given to enhancing the mental coping skill of those populations in quarantine and isolation to improve their psychological level.

The study further demonstrates that the mean total score of behavioral aggression was found to be 2.45±5.90 and about one-fourth (20.4%) of the participants scored above the mean of overt aggression scale. This finding shows COVID-19 stress was considerably related to physical and psychological aggression. This finding was consistent with a study done in the USA [14]. Following this, being male, having a low level of knowledge about COVID-19, and substance use were significantly and positively associated with mean overt aggression score. The relationship between substance use and aggression is complex and is indicative rather than conclusive [40, 41].

## Limitations of the study

The study has some limitations. Considering the limited availability of resources and the urgent or alarming effect of the COVID-19 pandemic outbreak, we implemented the convenience sampling technique. The present study used a cross-sectional design and relied on participant self-report.

## Conclusion

The current study revealed that suicidal behavior and aggression were prevalent among suspected cases of COVID-19..Female gender, having common mental disorders, manifesting the symptoms of COVID-19, and poor social support were significantly associated with suicide whereas male gender, knowledge about COVID-19, and substance use were significantly and positively associated with mean overt aggression score. Therefore, this study would help in developing targeted mental well-being strategies for those who are in need. Policymakers and helping professionals are advised that suicide behaviors are alarmingly common during the COVID-19 pandemic and vary based on gender, psychopathological and psychosocial factors.

## Supporting information

**S1 File. Data set used to determine the suicidal and aggressive behavior among populations within institutional quarantine and isolation centers of COVID-19 in eastern Ethiopia.**
(SAV)

## Acknowledgments

We are grateful to East Hararghe and Harari region COVID-19 task force. We also have great thanks to the study participants and data collectors.

## Author Contributions

**Conceptualization:** Tadesse Misgana, Dejene Tesfaye, Mandaras Tariku, Tilahun Ali, Daniel Alemu, Yadeta Dessie.

**Data curation:** Tadesse Misgana, Dejene Tesfaye, Tilahun Ali, Daniel Alemu, Yadeta Dessie.

**Formal analysis:** Tadesse Misgana, Mandaras Tariku, Tilahun Ali, Daniel Alemu.

**Funding acquisition:** Tadesse Misgana.

**Investigation:** Tadesse Misgana.

**Methodology:** Tadesse Misgana, Dejene Tesfaye, Mandaras Tariku.

**Project administration:** Tadesse Misgana, Mandaras Tariku, Yadeta Dessie.

**Resources:** Tadesse Misgana, Tilahun Ali, Yadeta Dessie.

**Software:** Tadesse Misgana, Tilahun Ali.

**Supervision:** Tadesse Misgana, Yadeta Dessie.

**Validation:** Tadesse Misgana, Dejene Tesfaye.

**Visualization:** Tadesse Misgana, Daniel Alemu.

**Writing – original draft:** Tadesse Misgana, Dejene Tesfaye, Mandaras Tariku, Tilahun Ali, Daniel Alemu, Yadeta Dessie.

**Writing – review & editing:** Tadesse Misgana, Dejene Tesfaye, Mandaras Tariku, Tilahun Ali, Daniel Alemu, Yadeta Dessie.

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
