## [Decision Letter · Decision Letter 0]

8 Sep 2022

PONE-D-22-11376Suicidal and aggressive behavior among populations within institutional quarantine and isolation centers of COVID-19 in eastern Ethiopia: A cross-sectional study

PLOS ONE

Dear Dr. Misgana,

Thank you for submitting your manuscript to PLOS ONE. After careful consideration, we feel that it has merit but does not fully meet PLOS ONE’s publication criteria as it currently stands. Therefore, we invite you to submit a revised version of the manuscript that addresses the points raised during the review process.

Your manuscript has been assessed by three reviewers whose reports can be found below. As you will see from the comments, the reviewers have raised a number of concerns that need attention. They request additional information on methodological aspects (e.g the sampling procedures) of the study and revisions to the statistical analyses. Could you please carefully revise the manuscript to address all comments raised?

We look forward to receiving your revised manuscript.

Kind regards,

Katrien Janin

Staff Editor

PLOS ONE

Journal Requirements:

- https://www.frontiersin.org/articles/10.3389/fpsyt.2021.753383/full

- https://www.medrxiv.org/content/10.1101/2020.07.09.20150136v1.full

- https://www.dovepress.com/front_end/suicide-and-suicidal-behaviors-in-the-context-of-covid-19-pandemic-in--peer-reviewed-fulltext-article-PRBM

In your revision ensure you cite all your sources (including your own works), and quote or rephrase any duplicated text outside the methods section. Further consideration is dependent on these concerns being addressed

"We thank Haramaya University for its financial support and for facilitating the research. Authors would also be grateful to study participants and the data collectors"

Reviewers' comments:

Reviewer's Responses to Questions

**Comments to the Author**

1. Is the manuscript technically sound, and do the data support the conclusions?

Reviewer #1: No

Reviewer #2: Partly

Reviewer #3: Partly

2. Has the statistical analysis been performed appropriately and rigorously? 

Reviewer #1: Yes

Reviewer #2: No

Reviewer #3: Yes

3. Have the authors made all data underlying the findings in their manuscript fully available?

Reviewer #1: Yes

Reviewer #2: No

Reviewer #3: No

4. Is the manuscript presented in an intelligible fashion and written in standard English?

Reviewer #1: Yes

Reviewer #2: Yes

Reviewer #3: No

5. Review Comments to the Author

Reviewer #1: Dear authors, thank you for the opportunity to read your work. This paper exploits a cross-sectional survey to investigate the prevalence of suicidal behavior and behavioral aggression among suspected cases of COVID-19 in Ethiopia. Descriptive and analytical statistics, and regression analyses were adopted to examine the results. However, at present, there are some critical issues to be addressed.

Major comments

Introduction:

1- The introduction is long. I suggest to summarize this section.

Method:

1- The introduction of the source population is long, I suggest to summarize it.

2- According to the type of variables studied, it is better to accurately mention the age range as the inclusion criteria..

3- The formula used to calculate the sample size is specified only for descriptive studies, but this study is also analytical and it is necessary to use related formulas. Therefore, according to the study, please calculate the power of the study.

4- The sampling method is described convenience method. Considering that prevalence is also mentioned in this study, it is necessary to have a random sampling so that the sample be representative of the source population and the results can be generalized to the population.

5- Considering that the result of the study in people who have mental illness can affect the overall results of the study, what explanation do you offer in this context?

Results:

1- Considering that the study was conducted in the quarantine and isolation center, what do you mean by symptom of covid-19, which was answered with yes and no? Please explain in this regard.

Discussion and conclusion:

1- It is better to include the important results of the study in the first paragraph and explain the main result.

The conclusion of the study should be consistent with the results of the study. In the second line of the conclusion section, it is said that the suicidal behavior and aggression has become more after the pandemic than before, while there was no documentation in this regard before the pandemic in the study.

Reviewer #2: This is an interesting study albeit with a relatively small and potentially biased sample. However, the study is important and was conducted with a population that is rarely included in public mental health studies.

Little information is offered about Sampling procedures. Authors assert they used a multi-centered cross-sectional design. However, there is no explanation how they chose the different centers, how many of them were included, etc. Nor is there an explanation how participants were selected.

Part of the weaknesses of the study are attributable to the fact that the sample size estimation was carried for a prevalence study, which clearly over-estimated the actual prevalence of suicidal behavior. However, the main issue is that the study also pursued some analytical component (e.g., identifying factors associated with suicidal behavior) but the sample size calculations did not contemplate this additional aim. The consequence is that several variables have relatively too low numbers in some cells, rendering association estimations quite unprecise. Lack of association in some of the variables and categories may be explained by lack of statistical power…

In line 230, the use of mean for age is not adequate as more than 60% of participants were < 30 years

In line 232, the last phrase should read “and 39 (9.9%) indicated they could not read and write (Table 1).”

I believe the authors would help readers understand better their sample if they wrote not only about cental tendency measures but also depicted the heterogeneity of their participants.

In line 243, the verb tense for have should be in the past.

In line 244 it is difficult to understand what “wear hears about COVID-19” means.

Line 268 does not match the figures in Table 3. If 86.5% reported use of Khat, then it follows that more than 22.7% had used at least one substance.

Line 299, eliminate repeated parenthesis sign.

Table 4 provides details about the aggression score by types of aggression. However, in the Methods section there is no mention of these subscales. Evidence of reliability and validity would be useful.

In Table 5, the acronyms used in the column labels are not clear. Please define them (e.g., COR, AOR). In addition, for the sake of clarity, please use the reference category systematically as the first or last category for each variable.

In Table 6 some of the p-values do not make sense. For example, in Educational status, those unable to write and read had a coefficient of 2.85 ((5% CI= 0.20, 5.51); yet, the p-value=0.22. This can’t be right. Similarly, please revise the figures for Primary school. Also, the number of significant digits should be similar for all figures, and there are some rounding errors in the narrative when reporting data from the table. Also, some causal language is used inappropriately when describing the association between substance use and aggression. The average score of aggression was higher for those who used substances, but that does not necessarily mean that the substance used caused the increase in aggression score.

As for the analysis of the data, it is not clear if the authors accommodated the complexity of recruiting survey participants from multiple centers. This may be problematic because the assumption of independence is violated.

In the Discussion section, as well as in the Abstract, authors assert their study show increased prevalence of suicidal behavior. I do not think this a correct way of conveying their main findings. The study shows a lower rate of suicide attempt than the meta-analyses they report. And even if the prevalence in the present study was higher they authors could not assert that there had been an increase in this specific population because they lack a baseline measure to compare results of the present study.

In lines 365, authors assert that females experience greater suicide burden than men. This is incorrect. The authors observed a higher prevalence of suicidal behaviors as measured by the particular instruments used for the study. However, the burden of suicide cannot be derived directly from their measure because it does not account for the actual proportion of suicide cases compared to attempts. The international literature shows a large difference in prevalence of complete suicide by males compared to females.

The phrase in lines 383-384 is not clear.

In lines 391-396, referring to social support, it seems more qualitative research is needed in this area. For example, community responses to COVID may only be understandable by using intensive dialogic research methods, as the interpretation and meaning of signs and symptoms may be an important mediator of behaviors.

In the Conclusions, the statement on lines 415-419 does not stand, as discussed earlier.

Some information from some references (e.g., 12, 13) seems to be missing. Probably the reference software used altered the original

Reviewer #3: The reported study is currently relevant in the context of the covid-19 pandemic; however, the data provided is a bit late since their collection was in November-December 2020. Even so, It is a significant contribution and usefulness.

The study is rigorous, although several aspects of the method need precision. They are listed below.

1. Regarding the procedure for choosing the sample, it would be necessary to clarify why the decision was made to do it for convenience instead of doing it randomly.

2. It is crucial to justify why the prevalence of suicidal behavior reported for the region before the pandemic was not taken. Instead, the authors decided to use the prevalence of suicidal behavior at 50%. While it is true that the prevalence in question did not exist during the pandemic, the decision may bias the size of the sample chosen.

3. Throughout the document, there are inconsistencies regarding the type of population studied. It is not clear if the participants were people suspected of being infected with covid or people who, after receiving positive results, were quarantined. For example, see lines 128-129; 134 to 138; 143-146 and; 345-346.

4. It would be desirable to expand the description of the data processing and analysis section regarding the number of variables that were included in the logistic regression (bivariate and multivariate).

5. Clarify why it was necessary to carry out the forward translation procedure with the study instruments. Was it necessary to do an adaptation process? The study understood that the instruments were already adapted and with ideal psychometric properties for the target population.

On the other hand, the results section begins by describing the peripheral findings to the study's objective, distracting the reader. The section that reports the association between suicidal behavior, aggressive behavior, and the explanatory variables suggest placing footnotes for the abbreviations COR and AOR. It would also be helpful to organize the data according to the groups "with suicidal behavior" and "without suicidal behavior" and specify the fit value of the model in each of them and for each explanatory variable analyzed.

The discussion section began by highlighting the prevalence of thoughts and behaviors related to suicide. This is inconsistent with the stated objective and the variables defined in the method. It is recommended to focus the discussion on the relationship raised in the objective.

Finally, the conclusions concluded that there was an increase in suicidal behavior during Covid-19, a scope outside the study's objective. Additionally, there is talk of the need to provide mental health care to people who test positive for Covid-19, suspicious people, family members in quarantine, and medical personnel. It is recommended to make conclusions based on the results obtained, and the people studied.

6. PLOS authors have the option to publish the peer review history of their article (what does this mean?). If published, this will include your full peer review and any attached files.

Reviewer #1: No

Reviewer #2: **Yes: **Dr. Fernando A. Wagner

Reviewer #3: No

---

## [Author Response · Author response to Decision Letter 0]

24 Sep 2022

POINT-BY-POINT REBUTTAL LETTER

We were pleased to have an opportunity to revise our manuscript entitled “Suicidal and aggressive behavior among populations within institutional quarantine and isolation centers of COVID-19 in eastern Ethiopia: A cross-sectional study”. In the revised manuscript, we have carefully considered journal requirements, the editor's and reviewer's suggestions and comments and we have tried to address it accordingly. The editor’s and reviewer’s comments were very helpful overall, and we are appreciative of such constructive feedback on our original submission. After addressing the issues raised, we feel the quality of the paper is much improved. 

Sincerely,

On behalf of all authors, 

Tadesse Misgana

Journal Requirements:

Authors’ response: We have checked the templates and made the adjustments to meet the journal requirements. Some of them are:

• We corrected all major sections (Abstract, Introduction, Materials and Methods, Results, Discussion) to level 1 heading, bold type, 18pt font, and sentence case 

• We corrected sub-sections of major sections to Level 2 heading, bold type, 16pt font, and sentence case. 

• We corrected sub-sections of within level 2 headings to level 3 heading, bold type, 14pt font, and sentence case. 

We hope that it now fits the style requirements, as described in the referred templates.

- https://www.frontiersin.org/articles/10.3389/fpsyt.2021.753383/full

- https://www.medrxiv.org/content/10.1101/2020.07.09.20150136v1.full

- https://www.dovepress.com/front_end/suicide-and-suicidal-behaviors-in-the-context-of-covid-19-pandemic-in--peer-reviewed-fulltext-article-PRBM

In your revision ensure you cite all your sources (including your own works), and quote or rephrase any duplicated text outside the methods section. Further consideration is dependent on these concerns being addressed.

Authors’ response: We have accepted the comments and we revised the manuscript and rephrased the duplicated text very thoroughly. Thanks. 

Authors’ response: we have tried to get the English language experts to review the manuscript for language usage, spelling, and grammar and now, we hope major language and grammar issues were fixed. Thank you

"We thank Haramaya University for its financial support and for facilitating the research. Authors would also be grateful to study participants and the data collectors"

Authors’ response: We have accepted the comments and we removed any funding-related text from the Acknowledgments section or other areas of the manuscript. We didn’t change the funding statement. Thank you.

Reviewers comment:

Reviewer #1: Dear authors, thank you for the opportunity to read your work. This paper exploits a cross-sectional survey to investigate the prevalence of suicidal behavior and behavioral aggression among suspected cases of COVID-19 in Ethiopia. Descriptive and analytical statistics, and regression analyses were adopted to examine the results. However, at present, there are some critical issues to be addressed.

Major comments

Introduction:

1- The introduction is long. I suggest to summarize this section.

Authors’ response: We have accepted the comments and we tried to summarize the introduction section as suggested. Thank you.

Method:

1- The introduction of the source population is long, I suggest to summarize it.

Authors’ response: We have accepted the comment and we have tried to summarize the introduction of the source population as suggested. Thank you.

2- According to the type of variables studied, it is better to accurately mention the age range as the inclusion criteria.

Authors’ response: We have accepted the comment and we have mentioned it as “aged 18 and above” in the revised manuscript. Thank you.

3- The formula used to calculate the sample size is specified only for descriptive studies, but this study is also analytical and it is necessary to use related formulas. Therefore, according to the study, please calculate the power of the study.

Authors’ response: At the beginning of the study, we have calculated the sample size for each objective. But, we reported only the one that gives us the larger sample size. In the revised manuscript, we have included all the calculated sample size including for the analytical study. Thank you. 

4- The sampling method is described convenience method. Considering that prevalence is also mentioned in this study, it is necessary to have a random sampling so that the sample be representative of the source population and the results can be generalized to the population.

Authors’ response: We randomly selected the sample quarantine and isolation centers from those established in the area. But, at the time of data collection, because of the pandemic was highly contagious and the number of cases dramatically increased in study area, we couldn’t get lists of suspected cases in those centers since there is no documentation. This forced us to take the sample conveniently.

5- Considering that the result of the study in people who have mental illness can affect the overall results of the study, what explanation do you offer in this context?

Authors’ response: Unique ethical considerations in research involving individuals with mental illness derive from the nature and heterogeneity of mental disorders. These unique considerations include the varying impact mental disorders can have on behavior and cognitive function, the possible effects of mental illness on decisional capacity, the stigma associated with mental illness, and the diminished social opportunities and political voice of many individuals with mental illness in our society. In this context, we have tried to develop valid and reliable methods of assessing the relevant decision-making abilities of people with mental illnesses and other disorders that may impair decisional capacity. Other than this, we have excluded those patients with severe mental illnesses that make them unable to give required information. Thank you

Results:

1- Considering that the study was conducted in the quarantine and isolation center, what do you mean by symptom of covid-19, which was answered with yes and no? Please explain in this regard.

Authors’ response: There were designated quarantine centers for suspected cases awaiting test results, but some were started to manifest the common symptoms of COVID-19 before they confirmed to have COVID-19 using lab test. This is accounted for 27% of the participants. 

Discussion and conclusion:

1- It is better to include the important results of the study in the first paragraph and explain the main result.

Authors’ response: We have accepted the comment and we included important results of the study in the first paragraph with explanations. Thank you. 

The conclusion of the study should be consistent with the results of the study. In the second line of the conclusion section, it is said that the suicidal behavior and aggression has become more after the pandemic than before, while there was no documentation in this regard before the pandemic in the study.

Authors’ response: We have accepted the comment and we make conclusions based on the results obtained as suggested. Thank you. 

Reviewer #2: This is an interesting study albeit with a relatively small and potentially biased sample. However, the study is important and was conducted with a population that is rarely included in public mental health studies.

Little information is offered about sampling procedures. Authors assert they used a multi-centered cross-sectional design. However, there is no explanation how they chose the different centers, how many of them were included, etc. Nor is there an explanation how participants were selected.

Authors’ response: We have accepted the comment and we added more about the sampling procedure, how to chose the centers and how the participants were selected as “From two isolation centers and 8 quarantine centers in Harari regional state, we randomly selected one isolation center and three quarantine centers. Again, the second area, East Hararghe zone of Oromia regional state had one isolation center and 12 quarantine center, and we randomly choose 4 of them. Finally, the participants were conveniently selected from each centers proportionally based on the number of suspected cases they occupied. The data were collected through face-to-face and self-reporting”. We included this on the revised manuscript.

Parts of the weaknesses of the study are attributable to the fact that the sample size estimation was carried for a prevalence study, which clearly over-estimated the actual prevalence of suicidal behavior. However, the main issue is that the study also pursued some analytical component (e.g., identifying factors associated with suicidal behavior) but the sample size calculations did not contemplate this additional aim. The consequence is that several variables have relatively too low numbers in some cells, rendering association estimations quite unprecise. Lack of association in some of the variables and categories may be explained by lack of statistical power…

Authors’ response: At the beginning, we have calculated the sample size for each objective. But, we reported only the one that gives us the larger sample size. In the revised manuscript, we have included all the calculated sample size including for analytical study. Thank you.

In line 230, the use of mean for age is not adequate as more than 60% of participants were < 30 years

Authors’ response: Since the age is not normally distributed (not adequate), we calculated the median for age with inter-quartile range as suggested. Thank you. 

In line 232, the last phrase should read “and 39 (9.9%) indicated they could not read and write (Table 1).”

Authors’ response: We have accepted and corrected the comment as recommended. Thank you.

I believe the authors would help readers understand better their sample if they wrote not only about cental tendency measures but also depicted the heterogeneity of their participants.

In line 243, the verb tense for have should be in the past.

In line 244 it is difficult to understand what “wear hears about COVID-19” means.

Authors’ response: We have accepted and corrected the above two comments. Thank you. 

Line 268 does not match the figures in Table 3. If 86.5% reported use of Khat, then it follows that more than 22.7% had used at least one substance.

Authors’ response: We have accepted and corrected the comments as “About 16.5%, 86.5%, and 16.5% of the respondents were reporting that they used Tobacco, Khat, and Alcohol in the previous three months respectively” in the revised manuscript. Thank you.

Line 299, eliminate repeated parenthesis sign.

Authors’ response: We have corrected the errors made.

Table 4 provides details about the aggression score by types of aggression. However, in the Methods section there is no mention of these subscales. Evidence of reliability and validity would be useful.

Authors’ response: we have accepted the comments and added more about sub scales of Modified Overt Aggression Scale (MOAS) in “Data collection instruments” sub-section of methods. Thank you. 

In Table 5, the acronyms used in the column labels are not clear. Please define them (e.g., COR, AOR). In addition, for the sake of clarity, please use the reference category systematically as the first or last category for each variable.

Authors’ response: We have accepted and addressed the comments in the revised manuscript. Thank you

In Table 6 some of the p-values do not make sense. For example, in Educational status, those unable to write and read had a coefficient of 2.85 ((5% CI= 0.20, 5.51); yet, the p-value=0.22. This can’t be right. Similarly, please revise the figures for Primary school. Also, the number of significant digits should be similar for all figures, and there are some rounding errors in the narrative when reporting data from the table. Also, some causal language is used inappropriately when describing the association between substance use and aggression. The average score of aggression was higher for those who used substances, but that does not necessarily mean that the substance used caused the increase in aggression score.

Authors’ response: We have accepted the comments and corrected the errors made. We also tried to rewrite interpretation of the data. Thank you. 

As for the analysis of the data, it is not clear if the authors accommodated the complexity of recruiting survey participants from multiple centers. This may be problematic because the assumption of independence is violated.

Authors’ response: we randomly selected the participants from multiple isolation and quarantine centers. Again, we have checked this assumption by creating a plot of residuals against time (i.e. the order of the observations) and we observed whether or not there is a random pattern. The model fit this assumption. Thank you.

In the Discussion section, as well as in the Abstract, authors assert their study show increased prevalence of suicidal behavior. I do not think this a correct way of conveying their main findings. The study shows a lower rate of suicide attempt than the meta-analyses they report. And even if the prevalence in the present study was higher they authors could not assert that there had been an increase in this specific population because they lack a baseline measure to compare results of the present study.

Authors’ response: We have accepted the comment and we made the amendment on conveying the finding based on existing literatures .We also rewritten conclusions based on the results obtained as suggested in the revised version of the manuscript. Again, we rewritten the discussion part based on the literatures. Thank you. 

In lines 365, authors assert that females experience greater suicide burden than men. This is incorrect. The authors observed a higher prevalence of suicidal behaviors as measured by the particular instruments used for the study. However, the burden of suicide cannot be derived directly from their measure because it does not account for the actual proportion of suicide cases compared to attempts. The international literature shows a large difference in prevalence of complete suicide by males compared to females.

Authors’ response: We are convinced by the comments and we corrected it. Thank you. 

The phrase in lines 383-384 is not clear.

Authors’ response: we have accepted the comments and we rewrite the mentioned statements.

In lines 391-396, referring to social support, it seems more qualitative research is needed in this area. For example, community responses to COVID may only be understandable by using intensive dialogic research methods, as the interpretation and meaning of signs and symptoms may be an important mediator of behaviors.

Authors’ response: You are correct. But, since during the data collection period, the pandemic was highly contagious and the numbers of cases were dramatically increased in the study area, we couldn’t conduct qualitative study. Rather we assessed the support system of the participants using standard tool (Oslo social support scale).

In the Conclusions, the statement on lines 415-419 does not stand, as discussed earlier.

Authors’ response: We have accepted the comment and we make conclusions based on the results obtained as suggested. Thank you. 

Some information from some references (e.g., 12, 13) seems to be missing. Probably the reference software used altered the original

Authors’ response: We have reviewed and completed the mentioned references. Thank you. 

Reviewer #3: The reported study is currently relevant in the context of the covid-19 pandemic; however, the data provided is a bit late since their collection was in November-December 2020. Even so, It is a significant contribution and usefulness.

The study is rigorous, although several aspects of the method need precision. They are listed below.

1. Regarding the procedure for choosing the sample, it would be necessary to clarify why the decision was made to do it for convenience instead of doing it randomly.

Authors’ response: We randomly selected sample quarantine and isolation centers from those established in the area. But, at the time of data collection, the pandemic was highly contagious and the number of cases dramatically increased in study area, we are unable to take the participants randomly. This forced us to take the sample conveniently. Thank you.

2. It is crucial to justify why the prevalence of suicidal behavior reported for the region before the pandemic was not taken. Instead, the authors decided to use the prevalence of suicidal behavior at 50%. While it is true that the prevalence in question did not exist during the pandemic, the decision may bias the size of the sample chosen.

Authors’ response: You are correct. But, there is no study conducted on suicide both during and before the pandemic in the study area. For the prevalence study, we took 50% since there is no study conducted. In the revised manuscript, we have included all the calculated sample size including for analytical study. Thank you.

3. Throughout the document, there are inconsistencies regarding the type of population studied. It is not clear if the participants were people suspected of being infected with covid or people who, after receiving positive results, were quarantined. For example, see lines 128-129; 134 to 138; 143-146 and; 345-346.

Authors’ response: The participants were people suspected of being infected with COVID-19 and peoples infected with the pandemic. The aim of this study is to determine the influence of mass quarantine and isolation regardless of the status of COVID-19 result. Thank you. 

4. It would be desirable to expand the description of the data processing and analysis section regarding the number of variables that were included in the logistic regression (bivariate and multivariate).

Authors’ response: we have accepted the comments and we addressed it in the revised version of the manuscript. Thank you. 

5. Clarify why it was necessary to carry out the forward translation procedure with the study instruments. Was it necessary to do an adaptation process? The study understood that the instruments were already adapted and with ideal psychometric properties for the target population.

Authors’ response: Since the data collectors and participants couldn’t understand the English version of the instruments, it is necessary to translate to local language. But, forward translation is only for checking whether the instrument was correctly translated or not. 

On the other hand, the results section begins by describing the peripheral findings to the study's objective, distracting the reader. The section that reports the association between suicidal behavior, aggressive behavior, and the explanatory variables suggest placing footnotes for the abbreviations COR and AOR. It would also be helpful to organize the data according to the groups "with suicidal behavior" and "without suicidal behavior" and specify the fit value of the model in each of them and for each explanatory variable analyzed.

Authors’ response: We have accepted and addressed the comments in the revised manuscript. Thank you

The discussion section began by highlighting the prevalence of thoughts and behaviors related to suicide. This is inconsistent with the stated objective and the variables defined in the method. It is recommended to focus the discussion on the relationship raised in the objective.

Authors’ response: The study has two objectives. The first one is determining the prevalence of suicidal and aggressive behavior among suspected cases in quarantine and isolation centers. The second is identifying factors associated with them. 

Finally, the conclusions concluded that there was an increase in suicidal behavior during Covid-19, a scope outside the study's objective. Additionally, there is talk of the need to provide mental health care to people who test positive for Covid-19, suspicious people, family members in quarantine, and medical personnel. It is recommended to make conclusions based on the results obtained, and the people studied.

Authors’ response: We have accepted the comment and we make conclusions based on the results obtained as suggested. Thank you.

---

## [Decision Letter · Decision Letter 1]

4 Dec 2022

PONE-D-22-11376R1Suicidal and aggressive behavior among populations within institutional quarantine and isolation centers of COVID-19 in eastern Ethiopia: A cross-sectional studyPLOS ONE

Dear Dr. Misgana,

Thank you for submitting your manuscript to PLOS ONE. After careful consideration, we feel that it has merit but does not fully meet PLOS ONE’s publication criteria as it currently stands. Therefore, we invite you to submit a revised version of the manuscript that addresses the points raised during the review process.

1. Thoroughly edit your work2. Provide justification for using all subscales of the MOAS knowing that your sample was in isolation3. Present your tables in APA format

We look forward to receiving your revised manuscript.

Kind regards,

Habil Otanga, Ph.D

Academic Editor

PLOS ONE

Journal Requirements:

Additional Editor Comments:

1. Thoroughly edit your work as suggested in reviewers' comments

2. Provide justification for using all subscales of the Modified Overt Aggression Scale

3. Present your tables in APA format

Reviewers' comments:

Reviewer's Responses to Questions

**Comments to the Author**

1. If the authors have adequately addressed your comments raised in a previous round of review and you feel that this manuscript is now acceptable for publication, you may indicate that here to bypass the “Comments to the Author” section, enter your conflict of interest statement in the “Confidential to Editor” section, and submit your "Accept" recommendation.

Reviewer #1: (No Response)

Reviewer #4: (No Response)

2. Is the manuscript technically sound, and do the data support the conclusions?

Reviewer #1: (No Response)

Reviewer #4: Partly

3. Has the statistical analysis been performed appropriately and rigorously? 

Reviewer #1: (No Response)

Reviewer #4: Yes

4. Have the authors made all data underlying the findings in their manuscript fully available?

Reviewer #1: (No Response)

Reviewer #4: Yes

5. Is the manuscript presented in an intelligible fashion and written in standard English?

Reviewer #1: (No Response)

Reviewer #4: Yes

6. Review Comments to the Author

Reviewer #1: (No Response)

Reviewer #4: 1. Language: Use and editing

Your work is full of words that should not be used where they are. You need to make the necessary adjustments. See below;

Line 68: replace the word "conceded"

Line 78: be clear on what you mean by "socio-economic crisis"

Line 81: What does "the other recognized precipitating risk factors of suicide could be negatively affected" mean?

Line 125 (in Methods): What is the outcome variable?

Line 136: replace the word "choose"

Line 138: What does "data were collected through face-to-face and self-reporting" mean?

Line 248-9: capital letters mid-sentence

2. Instruments

The MOAS has 4 categories. What justification exists for using ALL sub-scales of the scale especially auto and property aggression for people in isolation?

3. Pilot

Comment on why a pre-test was done.

4. Analysis

a. Why was data on substance use collected given that it referred to 3 months pre-isolation? Of what benefit was/is the data to the study? What objective does the data respond to?

b. Prepare table in APA format.

7. PLOS authors have the option to publish the peer review history of their article (what does this mean?). If published, this will include your full peer review and any attached files.

Reviewer #1: No

Reviewer #4: No

---

## [Author Response · Author response to Decision Letter 1]

21 Dec 2022

We were pleased to have an opportunity to revise our manuscript entitled “Suicidal and aggressive behavior among populations within institutional quarantine and isolation centers of COVID-19 in eastern Ethiopia: A cross-sectional study” for the second time. In the revised manuscript, we have carefully considered journal requirements, the academic editor's and reviewer's suggestions and comments and we have tried to address it accordingly. The academic editor’s and reviewer’s comments were very helpful overall, and we are appreciative of such constructive feedback on our original submission. After addressing the issues raised, we feel the quality of the paper is much improved. 

Sincerely,

On behalf of all authors, 

Tadesse Misgana

Journal Requirements:

Author’s response: We have checked all the references and it is complete and correct. We didn’t use the papers that have been retracted. Thank you. 

Editor Comments:

1. Thoroughly edit your work as suggested in reviewers' comments

Author’s response: We seriously took all the comments raised by the editor’s and reviewers, and we thoroughly edit the manuscript as suggested. Thank you. 

2. Provide justification for using all sub-scales of the Modified Overt Aggression Scale

Author’s response: Dear editor and reviewer, thank you for your concern. Some of the suspected individuals who were in isolation centers show all the types of aggression including aggression toward the self and properties. For instance, scratches skin, pulls hair out, hits self, hits fists into walls, throws self onto floor; Inflicts minor cuts, bruises, burns, or welts on self, and inflicts major injury on self, which are the manifestation of auto aggression. Again others showed aggression towards property like slamming door, rip clothing, urinates on floor; throwing objects down, kicks furniture, defaces walls; Breaks objects, smashes windows; and Sets fires. Because of such behaviors, we decided to assess all the 4 sub scales of MOAS. Thank you. 

3. Present your tables in APA format

Author’s response: We have accepted the comment and we presented the tables in APA format in the revised manuscript. Thank you. 

Reviewer’s comment:

Reviewer #4: 

1. Language: Use and editing

Your work is full of words that should not be used where they are. You need to make the necessary adjustments. See below;

Line 68: replace the word "conceded"

Line 78: be clear on what you mean by "socio-economic crisis"

Line 81: What does "the other recognized precipitating risk factors of suicide could be negatively affected" mean?

Line 125 (in Methods): What is the outcome variable?

Line 136: replace the word "choose"

Line 138: What does "data were collected through face-to-face and self-reporting" mean?

Line 248-9: capital letters mid-sentence

Author’s response: We have accepted all the comments raised regarding language use and editing, and we amended our manuscript as highlighted in the revised version. Thank you so much. 

2. Instruments

The MOAS has 4 categories. What justification exists for using ALL sub-scales of the scale especially auto and property aggression for people in isolation?

Author’s response: Some of the suspected individuals who were in isolation centers show all the types of aggression. For instance, they scratches skin, pulls hair out, hits self, hits fists into walls, throws self onto floor; Inflicts minor cuts, bruises, burns, or welts on self, and inflicts major injury on self which are the manifestation of auto aggression. Again others showed aggression towards property like slamming door, rip clothing, urinates on floor; throwing objects down, kicks furniture, defaces walls; Breaks objects, smashes windows; and Sets fires. Because of such behaviors, we decided to assess all the 4 sub scales of MOAS. Thank you. 

3.Pilot

Comment on why a pre-test was done.

Author’s response: We did the pre test in order to evaluate the acceptability and applicability of the procedures and tools. We have added this reason in the revised manuscript. Thank you.

4. Analysis

a. Why was data on substance use collected given that it referred to 3 months pre-isolation? Of what benefit was/is the data to the study? What objective does the data respond to?

Author’s response: We used a standardized WHO screening questionnaire Alcohol, Smoking, and Substance Involvement Screening Test (ASSIST) which define the current substance user as “When participants used at least one specified substance (for non medical purposes) in the last three months” and Ever substance users as “When participants used specified substance (for non-medical purposes) even once in their lifetime”. We assessed the substance use as independent variables since it known to be the risk factors for both suicidal and aggressive behavior. Thank you. 

b. Prepare table in APA format.

Author’s response: We have accepted the comment and we presented the tables in APA format. Thank you.

---

## [Decision Letter · Decision Letter 2]

12 Jun 2023

Suicidal and aggressive behavior among populations within institutional quarantine and isolation centers of COVID-19 in eastern Ethiopia: A cross-sectional study

PONE-D-22-11376R2

Dear Dr. Tadesse Misgana,

We’re pleased to inform you that your manuscript has been judged scientifically suitable for publication and will be formally accepted for publication once it meets all outstanding technical requirements.

Kind regards,

Habil Otanga, Ph.D

Academic Editor

PLOS ONE

Additional Editor Comments (optional):

Reviewers' comments:

Reviewer's Responses to Questions

**Comments to the Author**

1. If the authors have adequately addressed your comments raised in a previous round of review and you feel that this manuscript is now acceptable for publication, you may indicate that here to bypass the “Comments to the Author” section, enter your conflict of interest statement in the “Confidential to Editor” section, and submit your "Accept" recommendation.

Reviewer #1: All comments have been addressed

Reviewer #4: All comments have been addressed

2. Is the manuscript technically sound, and do the data support the conclusions?

Reviewer #1: Yes

Reviewer #4: Yes

3. Has the statistical analysis been performed appropriately and rigorously? 

Reviewer #1: Yes

Reviewer #4: Yes

4. Have the authors made all data underlying the findings in their manuscript fully available?

Reviewer #1: Yes

Reviewer #4: Yes

5. Is the manuscript presented in an intelligible fashion and written in standard English?

Reviewer #1: Yes

Reviewer #4: Yes

6. Review Comments to the Author

Reviewer #1: (No Response)

Reviewer #4: (No Response)

7. PLOS authors have the option to publish the peer review history of their article (what does this mean?). If published, this will include your full peer review and any attached files.

Reviewer #1: No

Reviewer #4: No
